# Resource Utilization of Residual Organic Sludge Generated from Bioenergy Facilities Using *Hermetia illucens* Larvae

**DOI:** 10.3390/insects15070541

**Published:** 2024-07-18

**Authors:** Kyu-Shik Lee, Eun-Young Yun, Tae-Won Goo

**Affiliations:** 1Department of Pharmacology, College of Medicine, Dongguk University, Gyeongju 38766, Republic of Korea; there1@dongguk.ac.kr; 2Department of Integrative Bio-Industrial Engineering, Sejong University, Seoul 05006, Republic of Korea; yuney@sejong.ac.kr; 3Department of Biochemistry, College of Medicine, Dongguk University, Gyeongju 38766, Republic of Korea

**Keywords:** organic sludge, livestock manure, *Hermetia illucens* larvae, fertilizer

## Abstract

**Simple Summary:**

Simple Summary: We attempted to establish a strategy for environmentally friendly treatment and resource utilization of residual organic sludge generated from bioenergy facilities (BF-rOS) using *Hermetia illucens* (black soldier fly) larvae (BSFL). Mixing BF-rOS with food waste (FW) was necessary for its environmentally friendly treatment and utilization. Furthermore, BSFL reared with the BF-rOS/FW mixture could be used as ingredients in animal feed. Therefore, this study demonstrates that the preparation of BF-rOS/FW mixtures is an effective strategy for the environmentally friendly treatment and resource utilization of BF-rOS using BSFL.

**Abstract:**

Residual organic sludge generated from bioenergy facilities (BF-rOS) is often disposed instead of recycled, thus contributing to further environmental pollution. This study explored the resource utilization of BF-rOS using *Hermetia illucens* larvae (BSFL). When BF-rOS was fed to BSFL for two weeks, the dry weight per individual BSFL was approximately 15% of that of BSFL that were fed food waste (FW). However, the dry weight increased by approximately two-fold in BSFL that were fed effective microorganism (EM)-supplemented BF-rOS containing 60% moisture. However, under both conditions, the BSFL did not mature into pupae. In contrast, the highest dry weight per BSFL was observed with the BF-rOS/FW (50%:50%) mixture, regardless of EM supplementation. Furthermore, the highest bioconversion rate was observed when the BSFL were fed the BF-rOS/FW (50%:50%) mixture, and the frass produced by the BSFL contained fertilizer-appropriate components. In addition, the nutritional components of the BSFL exhibited a nutrient profile suitable for animal feed, except for those fed BF-rOS only. In conclusion, this investigation demonstrates that BF-rOS should be recycled for fertilizer production by mixing it with FW as a BSFL feed, which generates the valuable insect biomass as potential nutrition for animal feeding.

## 1. Introduction

Organic waste, including livestock, human manure, and food waste (FW), is a major cause of environmental pollution, such as water and soil contamination and odor generation. Livestock manure is a major contributor to greenhouse gas emissions. According to the Food and Agriculture Organization of the United Nations reports, as of 2018, organic waste accounted for the emission of 1.4 billion tons of CO_2_ [1].

Global meat consumption is steadily increasing, leading to an increase in the number of livestock, which in turn causes deforestation, soil contamination, increased livestock manure generation, enhanced greenhouse gas emissions, and soil contamination [2,3,4]. In China, which accounts for over 30% of the global meat production, the chemical oxygen demand discharge ratio increased by 41.87% from 2007 to 2020 [5]. This implies a consistent increase in the generation of livestock manure owing to livestock breeding.

Bioenergy conversion and fertilization have been predominantly implemented to address the environmental pollution caused by increasing livestock manure production and valorization [6,7]. The most common method for valorizing livestock manure is fertilization. However, fertilized livestock manure is a substantial source of greenhouse gas emissions. Therefore, efforts are being made globally to steadily increase the proportion of bioenergy converted from livestock manure through anaerobic digestion to reduce greenhouse gas emissions [8,9].

Bioenergy conversion is considered an effective recycling strategy for livestock manure because it produces biogas, and the generated organic sludge can be used as a raw material for organic fertilizers [10]. Several reports have suggested that anaerobic co-digestion of livestock manure and FW is an excellent strategy for enhancing biogas production [11,12,13]. In addition, the liquid organic sludge generated during bioenergy production through the co-digestion of livestock manure and FW can be used as a valuable raw material for liquid fertilizers. However, a large amount of residual organic sludge remains after the production of liquid fertilizer raw materials from liquid organic sludge generated in bioenergy facilities, which is still being disposed of. Residual organic sludge generated from bioenergy facilities (BF-rOS) can cause water and air pollution, and serves as a transmission route for pathogens because it retains moisture, organic matter (such as protein, carbohydrate, and lipid), minerals, and pathogens [14,15,16,17]. However, strategies for the treatment of BF-rOS have not yet been established.

Black soldier fly (*Hermetia illucens*) larvae (BSFL) are well-known environmentally friendly insects that feed on various organic wastes, including livestock manure and food waste; they require low rearing costs and have a short life cycle, thus facilitating mass production [18,19,20]. Therefore, BSFL are considered suitable insects for resource recovery through the environmentally friendly treatment of organic waste such as livestock manure, food waste, and organic sludge [21,22,23]. Numerous investigations have demonstrated that BSFL can reduce the amount of organic waste through its bioconversion, and the frass obtained during bioconversion can be used as a raw material for organic fertilizers [24,25]. Furthermore, BSFL can be used as feed for livestock and aquaculture [26,27,28]. In addition, several studies have described the potential development of antibiotics from BSFL, as well as the feasibility of biodiesel production using the lipids of BSFL [18,23,29,30,31]. These results indicate that BSFL are suitable insects for the environmentally friendly treatment of organic waste and have a high potential for industrial applications. Therefore, this study proposes a resource recovery method based on the bioconversion of BF-rOS using BSFL. Furthermore, the feasibility of utilizing BSFL as feed for livestock and aquaculture was evaluated.

## 2. Materials and Methods

### 2.1. H. illucens Larvae

BSFL were obtained by Greenteko Corporation (Hwaseong, Republic of Korea) and were grown at 26 ± 1 °C under 60% relative humidity. Each BSFL group that was fed BF-rOS or BF-rOS/FW mixtures was grown for 14 days. The BF-rOS was produced in the form of solid residue with 80% moisture after anaerobic digestion of an 80:20 mixture of cow manure and FW for biogas production, followed by the extraction of liquid fertilizer from the resulting residue.

### 2.2. Moisture Adjustment of BF-rOS and Preparation of BF-rOS/FW Mixture

BF-rOS with 80% moisture and sterilized FW were provided from the Resource Recycling Agricultural Center of the Nonsan Livestock Cooperative Federation (Nonsan, Republic of Korea). BF-rOS with 70% and 60% moisture was prepared by adding sawdust to BF-rOS with 80% moisture. The BF-rOS/FW mixtures were manufactured by blending BF-rOS with 80% moisture and FW at conditioned ratios (90:10, 70:30, and 50:50), and the final moisture content was adjusted to 80% using sawdust.

### 2.3. BSFL Breeding

A thousand 6-day-old BSFL per replicate were inoculated into BF-rOS or BF-rOS/FW mixtures, reared for 14 days, and then harvested to evaluate the survival rate, dried weight of individuals, and suitability as feed. In addition, the growth of BSFL is known to be enhanced by an appropriate addition of effective microorganisms (EM); to assess the effect of EM on BSFL growth, BSFL were reared in BF-rOS or BF-rOS/FW mixtures with conditioned ratios of EM (0, 1.0, 1.5, and 2.0%) [30,32].

### 2.4. Analysis of Bioconversion Efficiency of BSFL

We analyzed the bioconversion efficiency of the BF-rOS and BF-rOS/FW mixtures with conditioned ratios of EM (0, 1.0, 1.5, and 2.0%, respectively). Residual matter was collected after growing the BSFL for 14 days, and the weight of the residual matter was measured after a four-week maturation. Then, the feed residual rate was calculated using the following formula: Feed residual rate = [weight of residue matter of sample/weight of residue matter of BF-rOS (EM 0%)] × 100 (%)}. Finally, bioconversion efficiency was expressed as the relative residual reduction of BF-rOS and was calculated using the following formula: Bioconversion efficiency = [feed residual rate of BF-rOS (EM 0%)—feed residual rate of sample]. Experiments were independently conducted in triplicate. Experiments were independently conducted in triplicate.

### 2.5. Analysis of Component Content of BSFL Frass

The contents of the BSFL frass generated from BF-rOS or BF-rOS/FW mixtures were analyzed to evaluate their suitability as organic fertilizer raw materials. BSFL frass was collected after growing the BSFL for 14 days, and the component contents were analyzed after a four-week maturation. The components of the BSFL frass were analyzed by Cheillab Corporation (Seoul, Republic of Korea), in accordance with the standards stipulated in the Fertilizer Control Act of the Republic of Korea. Heavy metals and NaCl were analyzed with an inductively coupled plasma method; capsaicin was analyzed using liquid chromatography tandem mass spectrometry; and the total nitrogen content was analyzed using a sulfuric acid digestion method. Hydrochloric acid-insoluble residue analysis was performed using the method described by Van Keulen and Young [33].

### 2.6. Analysis of Nutritional Compsition of BSFL

To assess BSFL as feed for livestock and aquaculture, the nutritional composition of BSFL was examined. The BSFL were harvested after growing for 14 days and dried in a microwave for 6 min. Analyses of crude protein, crude fat, crude fiber, crude ash, and moisture were conducted at Cheillab Corporation according to the standards prescribed in the Control Livestock and Fish Feed Act of the Republic of Korea. Crude protein was analyzed using the Kjeldahl method, crude fat was analyzed using an ether extraction method, crude fiber was determined using a filter paper method, and moisture was determined using a loss-on-drying method. Crude ash analysis was performed according to AOAC Official Method 942.05.

### 2.7. Analysis of Heavy Metals and Aflatoxins in BSFL

We assessed the contents of heavy metals and aflatoxins in the BSFL to verify their safety as feed for livestock and aquaculture. The BSFL harvested after breeding for 14 days were dried in a microwave oven for 6 min. The heavy metal and aflatoxin contents in BSFL were measured at Cheillab Corporation in accordance with the standards imposed in the Control Livestock and Fish Feed Act of the Republic of Korea. Aflatoxins were analyzed using liquid chromatography tandem mass spectrometry.

### 2.8. Evaluation of Cytotoxicity of BSFL

To verify the safety of BSFL as animal feed, their cytotoxic activity was assessed using the human intestinal cell line Caco-2 and the mouse fibroblast cell line L929. A thousand cells were plated into each well of a 96-well plate and allowed to attach for 24 h. Subsequently, the cells were treated with conditioned medium supplemented with various concentrations of BSFL powder (0, 1000, 3000, and 5000 μg/mL) for 24 h. After the treatment period, the conditioned medium was removed and cells were incubated with 3-(4,5-dimethylthiazol-2-yl)-2,5-diphenyltetrazolium bromide (MTT) reagent for 4 h in the dark. The formazan crystals formed by the reduction of MTT by viable cells were dissolved in dimethyl sulfoxide, and the absorbance of the solubilized formazan was measured at 540 nm using an Epoch 2 ELISA Reader (Agilent Technologies, Santa Clara, CA, USA).

### 2.9. Statistical Analysis

All data were statistically analyzed using one-way analysis of variance followed by Tukey’s post hoc test, using SPSS Ver. 20.0 (SPSS Inc., Chicago, IL, USA). Statistical significance was set at *p* values less than 0.05. Values in all data are displayed as means ± standard deviations.

## 3. Results

### 3.1. The Growth of BSFL When Fed BF-rOS

We reared BSFL from BF-rOS at 60%, 70%, and 80% moisture to evaluate the growth of BSFL in BF-rOS. The results showed that no difference in BSFL survival was observed between groups fed FW or BF-rOS with different moisture contents; the dry weight/larva of BSFL fed BF-rOS was significantly lower than that of the BSFL fed FW (Figure 1A). Additionally, the dry weight/larva of BSFL that were fed BF-rOS with 60% moisture was significantly increased by supplementing 1.0 with 1.5% EM. However, the increased dry weight/larva of BSFL fed BF-rOS with 60% moisture accounted for approximately 25% of the BSFL that were fed FW (Figure 1B). Therefore, these results demonstrate that BSFL cannot grow sufficiently when fed with BF-rOS.

### 3.2. The Growth of BF-rOS When Fed BF-rOS/FW Mixture

As shown Figure 1, the BSFL did not grow sufficiently when fed BF-rOS. Therefore, we investigated whether adjusting the ratio of BF-rOS to FW could enhance the growth of the BSFL. The results revealed that the highest dry weight/larva was observed in the BSFL when provided with a 50:50 BF-rOS/FW mixture, with no influence on survival (Figure 2). Furthermore, the improvement in BSFL growth by consuming the BF-rOS/FW mixture was closely correlated with an increase in the FW ratio. Moreover, the dry weight/larva of BSFL fed the 50:50 BF-rOS/FW mixture was higher than that of BSFL fed solely FW. However, supplementation of the mixture with EM did not affect the survival and growth of the BSFL (Figure 2). These findings suggest that the 50:50 BF-rOS/FW mixture is suitable for breeding BSFL.

### 3.3. Bioconversion Effciency of BF-rOS and BF-rOS/FW Mixtures by BSFL

Based on the aforementioned results, we estimated the bioconversion efficiency of BSFL fed BF-rOS or BF-rOS/FW mixtures. As shown in Table 1, the bioconversion efficiency of BSFL fed with BR-rOS alone decreased proportionally with moisture content. However, the higher reduction in bioconversion efficiency compared to the added sawdust suggests that while sawdust used for moisture adjustment may influence bioconversion efficiency, the efficiency is likely closely correlated with the moisture content in BF-rOS, unaffected by the presence of EM, when BSFL were fed solely BF-rOS (Table 1). In contrast, an increased bioconversion efficiency was noted when the BSFL were fed BF-rOS/FW mixtures compared to when they were fed BF-rOS alone, with the increase associated with the ratio of FW in the mixture despite the increased addition of sawdust for moisture adjustment. Furthermore, bioconversion efficiency was enhanced by supplementation with EM, except for the 70:30 BF-rOS/FW mixture (Table 1). These results demonstrate that the bioconversion efficiency of BF-rOS should be enhanced by a mixture of FW and EM supplementation.

### 3.4. Analysis of the Composition of BSFL Frass

We examined the composition of the frasses of the BSFL that were fed BF-rOS and BF-rOS/FW mixtures to evaluate their suitability as organic fertilizers. The results showed that all the BSFL frasses contained heavy metals below the threshold specified in the Fertilizer Control Act of the Republic of Korea (Table 2). Additionally, we verified that the levels of organic matter, moisture, hydrochloric acid-insoluble residue, and capsaicin, as well as the ratio of nitrogen to organic matter, met the required standards. However, the NaCl content of the frass produced from 100% BSFL fed FW exceeded the standard limit, rendering it unsuitable for use as a fertilizer (Table 2). These results suggest that all BSFL frass, except for the frass generated from 100% BSFL fed FW, are appropriate for use as organic fertilizer.

### 3.5. Evaluation of BSFL Composition as Feed Ingredients

We also analyzed the nutritional composition, heavy metals, and aflatoxins in BSFL that were fed BF-rOS or BF-rOS/FW mixtures to estimate their suitability as feed ingredients. The highest crude protein and ash levels were observed in the BSFL fed BF-rOS alone, which was accompanied by the lowest crude fat content (Table 3). Conversely, the highest crude fat content, along with the lowest crude ash content, was observed in the BSFL fed FW alone. In the case of BSFL fed with BF-rOS/FW mixtures, the crude fat and ash contents in BSFL correlated with the proportion of FW in the mixtures. As shown in Table 3, the increase in crude fat and the decrease in crude ash were associated with the proportion of FW in the mixtures. However, the EM content of the mixtures did not affect their nutritional composition.

In the heavy metal content analysis, As was not detected in any of the BSFL samples. The highest Cd content was observed in the BSFL that were fed FW, along with the lowest Cr content (Table 4). In contrast, the Cr content was the highest in BSFL that were fed BF-rOS alone. Additionally, the Cr content in BSFL fed BF-rOS/FW mixtures was closely linked to the proportion of FW in the mixtures.

In addition, we analyzed aflatoxin levels in the BSFL that were fed BF-rOS or BF-rOS/FW mixtures. No aflatoxins were detected in the BSFL. This result demonstrates that the growth and processing of BSFL were conducted safely without exposure to contaminants such as fungi and molds.

### 3.6. Evaluation of Cytotoxicity of BSFL

The cytotoxicity of BSFL was evaluated against the human intestinal cell line Caco-2 and the mouse fibroblast cell line L929 to verify the safety of BSFL as animal feed additives. The results showed no decrease in cell survival in either Caco-2 or L929 cell lines when exposed to BSFL powder. These findings suggest that BSFL do not exhibit cytotoxic activity against normal cells. Consequently, BSFL appear safe for use as animal feed.

## 4. Discussion

In this study, we attempted to establish the conditions for the bioconversion of BF-rOS using BSFL. We investigated the suitability of the BSFL frass as a fertilizer and the appropriateness of BSFL as feed ingredients. As shown in Figure 1, the body weight/individual of BSFL fed BF-rOS alone for 14 days was only approximately 16.7% of that of BSFL fed FW. Conversely, the body weight/individual of BSFL fed on BF-rOS/FW mixtures increased proportionally with FW content (Figure 2). Notably, the body weight of BSFL fed a 50:50 BF-rOS/FW mixture increased by approximately 33% compared to that of BSFL fed FW alone. In a recent study, Kofroňová et al. suggested that biogas digestate is suitable feed for BFSL [34]. However, despite using biogas digestate as feed, obtained from wet fermentation that included both liquid and solid sludge, the growth of BSFL was lower than that of BSFL fed with FW. Furthermore, a report has shown that when digested organic sludge was supplied for approximately 14 days, the growth of BSFL was significantly lower than those fed with poultry manure or undigested sludge [35]. In this investigation, we used BF-rOS, a digested organic sludge as feed, from which liquid sludge had been removed, potentially resulting in lower nutrient content compared to previous studies. Consequently, these results indicate that BF-rOS alone does not provide sufficient nutrients for BSFL growth, necessitating mixing FW with BF-rOS for effective BSFL cultivation.

The failure of BSFL growth when fed BF-rOS alone suggests that BF-rOS cannot be used as a feed for BSFL, implying inadequate bioconversion under such conditions. Although the bioconversion efficiency in BF-rOS alone enhanced with increasing moisture content, the significantly lower body weight/individual of BSFL fed BF-rOS alone compared to those fed FW, regardless of moisture content, suggests that the nutritional content of the feed, rather than moisture content, is more crucial for BSFL growth (Figure 1 and Table 1). Actually, the bioconversion efficiency of the BSFL fed on BF-rOS/FW mixtures, except for the 90:10 BF-rOS/FW mixture without EM, was significantly higher than BSFL fed BF-rOS alone with 80% moisture (Table 1). Furthermore, the bioconversion efficiency enhanced with increasing FW content, with the highest bioconversion efficiency observed when the BSFL were fed a 50:50 BF-rOS/FW mixture containing 1.5% EM. These findings indicate that the preparation of BF-rOS/FW mixtures with an appropriate EM content is necessary for the effective bioconversion of BF-rOS using BSFL.

The resource utilization of bioconverted BF-rOS is crucial for reducing environmental pollution caused by BF-rOS and increasing its added value. Although the BSFL frass generated from BSFL fed BF-rOS alone contained acceptable levels of heavy metals and nutrients for use as an organic fertilizer, the bioconversion of BF-rOS by BSFL was insufficient (Table 1 and Table 2). In contrast, sufficient bioconversion was achieved with BSFL fed on FW alone, which was used as a control, but the resulting BSFL frass contained 5.33% NaCl, exceeding the acceptable threshold for organic fertilizer materials. However, the BSFL frass produced from BF-rOS/FW mixtures contained NaCl levels ranging from 1.20% to 1.88%, which were within acceptable limits. The NaCl content increased proportionally with the FW content in the BF-rOS/FW mixtures. Consequently, these results suggest that to ensure effective resource utilization of BF-rOS through bioconversion by BSFL, using FW with a low NaCl content in the mixture is advantageous to ensure safety.

BSFL are used as feed for livestock and aquaculture. Therefore, we assessed whether the BSFL obtained after the bioconversion of BF-rOS were suitable as animal feed. BSFL fed BF-rOS alone had the highest crude protein and ash contents at 49.89% and 18.93%, respectively, but significantly lower crude fat at 1.34% compared to those fed FW alone or BF-rOS/FW mixtures (Table 3). The crude fat content in BSFL fed BF-rOS/FW mixtures was inversely correlated with the BF-rOS content. Many investigations have shown that the nutritional composition of BSFL is determined by the nutritional composition of the feed [26,28,36,37,38]. In this investigation, BSFL fed with BF-rOS exhibit very low crude fat content. This implies that BF-rOS likely contains very low-fat content. In fact, there is a report indicating that crude fat content is not applicable in digested organic sludge [35]. This information suggests that the decreased growth and very low-fat content observed in BSFL fed BF-rOS alone may be attributed to deficiencies in the nutrients contained within BF-rOS. On the other hand, BSFL fed 70:30 and 50:50 BF-rOS/FW mixtures had optimal nutrient compositions suitable for animal feed. Safety assessments for heavy metals in all BSFL showed no detection of As, with Cd, Hg, and Cr levels below the safety thresholds (Table 4). Additionally, aflatoxins were not detected in any of the BSFL. Toxicity evaluation of animal cells revealed no cytotoxicity of any BSFL powder (Figure 3). These results indicate that BSFL fed BF-rOS/FW mixtures possess appropriate nutritional profiles and are safe feed ingredients.

In conclusion, this investigation suggests that for the environmentally friendly treatment and resource utilization of BF-rOS using BSFL, it is necessary to mix BF-rOS with FW in appropriate proportions.

## Figures and Tables

**Figure 1 insects-15-00541-f001:**
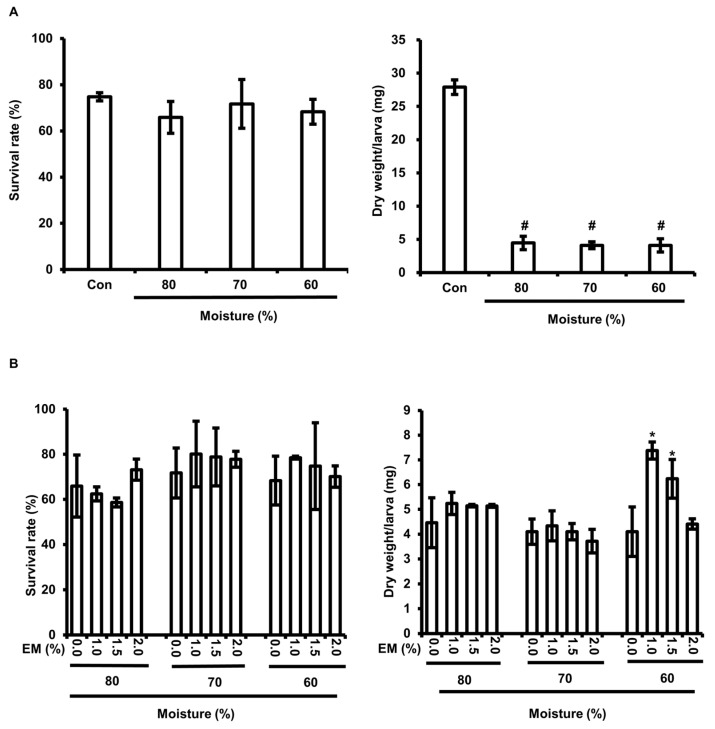
Effect of moisture in residual organic sludge generated from bioenergy facilities (BF-rOS) and effective microorganism (EM) on the survival and body weight of *Hermetia illucens* (black soldier fly larvae) (BSFL). (**A**) Six-day-old BSFL were inoculated into BF-rOS with varying moisture levels and allowed to grow for 14 days. Food waste (FW) served as a positive control. The statistical significance of each experimental group was determined by comparing it to the control. # indicates *p* < 0.0001. (**B**) Six-day-old BSFL were inoculated into BF-rOS with varying moisture and EM levels and allowed to grow for 14 days. The statistical significance of each experimental group was determined by comparing it to the control. * indicates *p* < 0.05. (**A**,**B**) Surviving BSFL were harvested and counted to determine survival rate. They were dried to measure the dry weight/larva. Dry weight/larva was calculated by dividing total dry weight of BSFL by the total number of BSFL. All experiments were conducted independently in triplicate. Values are presented as the mean ± standard deviation of survival rate or dry weight/larva.

**Figure 2 insects-15-00541-f002:**
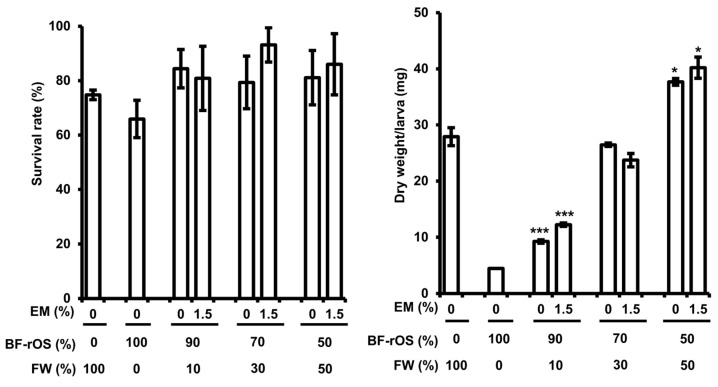
Effect of BF-rOS/FW mixtures and EM on the survival and body weight of BSFL. BSFL were grown for 14 days after inoculating 6-day-old BSFL, and then we collected and counted the number of surviving BSFL to determine survival rate. They were dried to measure the dry weight/larva. Dry weight/larva was determined by dividing total dry weight of BSFL by the total number of BSFL. All experiments were conducted independently in triplicate. Values are presented as the mean ± standard deviation of survival rate or dry weight/larva. The statistical significance of each experimental group was determined by comparing it to FW alone. * and *** indicate *p* < 0.05 and *p* < 0.001, respectively.

**Figure 3 insects-15-00541-f003:**
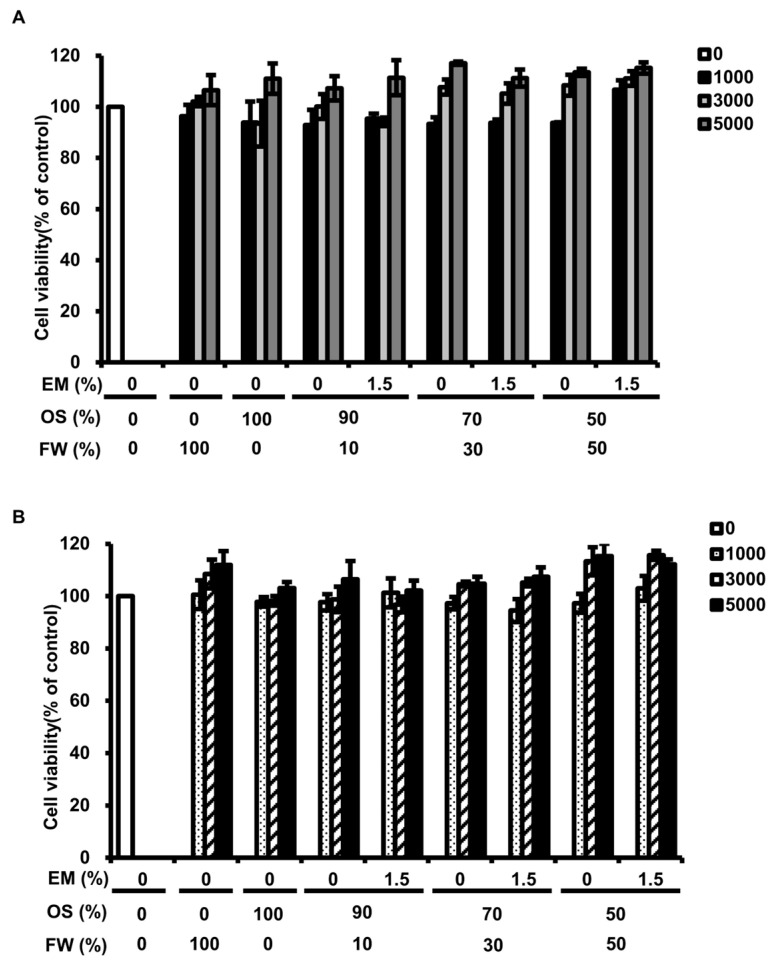
Effect of BSFL powder on animal cell survival. (**A**) Survival of human intestinal cell line Caco-2. (**B**) Survival of mouse fibroblast cell line L929. (**A**,**B**) Cell survival of BSFL powder-treated cells was determined as a percentage of the survival of untreated control cells. No significant differences were observed in any of the samples. OS indicates BF-rOS.

**Table 1 insects-15-00541-t001:** Bioconversion efficiencies of BF-rOS and BF-rOS/FW mixtures by BSFL.

Feed	EM (%)	Initial Weight of Feed (g)	Relative Residual Reduction (%)
BF-rOS (80%) ^1^	0.0	1000	0.00 ± 1.45
	1.0	1000	−10.30 ± 0.80 *
	1.5	1000	−9.62 ± 4.18 *
	2.0	1000	−5.76 ± 1.28
BF-rOS (70%)	0.0	1000	−74.08 ± 4.18 ***
	1.0	1000	−77.49 ± 1.90 ***
	1.5	1000	−74.53 ± 2.91 ***
	2.0	1000	−101.77 ± 2.91 ***
BF-rOS (60%)	0.0	1000	−157.38 ± 4.44 ***
	1.0	1000	−149.66 ± 2.54 ***
	1.5	1000	−153.74 ± 3.18 ***
	2.0	1000	−144.21 ± 6.40 ***
FW	0.0	1000	50.75 ± 2.59 ***
BF-rOS:FW (90:10) ^2^	0.0	1000	−1.45 ± 9.94
	1.5	1000	9.44 ± 0.95
BF-rOS:FW (70:30)	0.0	1000	20.79 ± 3.54 *
	1.5	1000	20.79 ± 6.08 *
BF-rOS:FW (50:50)	0.0	1000	36.22 ± 1.59 **
	1.5	1000	41.22 ± 0.32 **

^1^ Moisture in BF-rOS. ^2^ Mixture ratio. * *p* < 0.05 vs. BF-rOS (80%, EM 0%), ** *p* < 0.01 vs. BF-rOS (80%, EM 0%), *** *p* < 0.001 vs. BF-rOS (80%, EM 0%).

**Table 2 insects-15-00541-t002:** The compositional analysis of BSFL frass.

Composition	BF-rOS	BSFL Frass							
		Feed [Mixture Ratio of BF-rOS and FW (EM)]							
		100/0 (0.0)	0/100 (0.0)	90/10 (0.0)	90/10 (1.5)	70/30 (0.0)	70/30 (1.5)	50/50 (0.0)	50/50 (1.5)
OM (%)	69.63	74.55	71.72	75.82	72.71	73.98	75.55	72.67	72.47
As (mg/kg)	ND	ND	0.55	0.16	0.27	0.31	0.38	ND	0.37
Cd (mg/kg)	0.0063	0.0336	ND	0.034	0.064	0.035	0.062	0.077	0.11
Ag (mg/kg)	0.21	ND	ND	0.21	0.13	ND	ND	ND	ND
Pb (mg/Kg)	0.48	0.49	1.42	0.90	0.76	0.73	0.16	0.74	0.60
Cr (mg/kg)	7.67	6.26	7.10	6.10	6.72	6.44	5.88	7.35	7.36
Cu (mg/kg)	191.10	169.00	28.70	156.60	167.30	152.70	156.60	141.80	141.90
Ni (mg/kg)	7.49	6.02	1.86	5.56	5.64	5.56	6.49	5.39	5.59
Zn (mg/kg)	376.04	344.20	46.20	308.10	336.70	285.10	303.00	284.70	259.20
NaCl (%)	1.03	0.92	5.33	1.20	1.21	1.76	1.75	1.77	1.88
MO (%)	19.60	14.16	19.47	13.19	14.70	16.00	14.10	15.62	16.75
C/N (%)	33.80	38.42	35.68	42.60	35.82	42.27	41.74	36.52	38.34
HCl (%)	1.88	1.75	0.90	1.56	1.02	1.69	1.68	0.98	1.24
CAP (mg/kg)	ND	ND	0.031	ND	0.010	0.020	0.020	0.080	0.080

OM: Organic matter; As: Arsenic; Cd: Cadmium; Ag: Mercury; Pb: Lead; Cr: Chromium; Cu: Copper; Ni: Nickel; Zn: Zinc; NaCl: Salt; MO: Moisture; C/N: Ratio of nitrogen to organic matter; HCl: Hydrochloric acid-insoluble residue; CAP: Capsaicin; ND: Not detected.

**Table 3 insects-15-00541-t003:** Nutritional composition analysis of BSFL.

Nutritional Composition (%)	Feed [Mixture Ratio of BF-rOS and FW (EM)]							
	100/0 (0.0)	0/100 (0.0)	90/10 (0.0)	90/10 (1.5)	70/30 (0.0)	70/30 (1.5)	50/50 (0.0)	50/50 (1.5)
Crude protein	49.89	38.62	47.3	47.4	44.98	44.3	43.16	44.50
Crued fat	1.34	30.56	8.60	7.97	16.97	15.26	23.61	21.18
Crude fiber	9.71	7.88	9.67	9.61	8.69	8.53	7.97	8.46
Crude ash	18.93	8.38	15.37	15.99	12.73	12.39	11.74	12.39
Moisture	5.00	6.01	5.15	4.58	4.17	6.75	3.44	1.81

**Table 4 insects-15-00541-t004:** The analysis of heavy metals in BSFL.

Heavy Metal (mg/kg)	Feed [Composition Ratio of BF-rOS and FW (EM) in Each Feed]							
	100/0 (0.0)	0/100 (0.0)	90/10 (0.0)	90/10 (1.5)	70/30 (0.0)	70/30 (1.5)	50/50 (0.0)	50/50 (1.5)
As	ND	ND	ND	ND	ND	ND	ND	ND
Cd	0.22	0.42	0.18	0.15	0.21	0.24	0.20	0.26
Pb	1.19	1.33	1.25	0.94	0.80	0.74	0.82	1.00
Hg	0.055	0.100	0.057	0.043	0.050	0.050	0.047	0.058
Cr	3.88	1.99	3.46	3.26	3.03	2.79	2.41	2.46

## Data Availability

The data presented in this study are available on request from the corresponding author.

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
