# Peer review of "Resource Utilization of Residual Organic Sludge Generated from Bioenergy Facilities Using Hermetia illucens Larvae"

_insects, 2024, doi:10.3390/insects15070541_

Round 1

Reviewer 1 Report

Comments and Suggestions for Authors

This manuscript entitled “Resource utilization of residual organic sludge generated from bioenergy facilities using Hermetia illucens larvae” presents an important investigation of the bioconvertion of residual organic sludge generated from bioenergy facilities (BF-rOS) using black soldier fly larvae (BSFL) to generate the edible insect-sourced nutrition and organic fertilizers. The author not only conformed the BSFL could efficiently bioconvert the BF-rOS mixed with food waste (FW), further optimized the most efficient ratio of BF-rOS and FW mixtures, but also tried to evaluated the potential of insect biomass as animal feeding nutrition and frass as organic fertilizers. These findings provide a new efficient way to treat the BF-rOS for recycling this organic waste for fertilizer production and animal feeding. The manuscript content is logic, the results are clearly described, discussion is sufficient, and the writing is well readable. However, the deficiencies in method description, result analysis, figures, tables, are especially in the indicator adequacy to support the conclusion, were also found as described as followed:

1.    In most cases, the larvae of black soldier fly are abbreviated as “BSFL”, which is also suggested in this manuscript.

2.    Why selecting food wastes as the supplementary materials to improve the bioconversion of BF-rOS using black soldier fly larvae (BSFL)? Beside food wastes, many organic wastes, such as pig and chicken manures, are also good materials for BSFL conversion, so I suggest the authors to describe the reason for choosing FW.

3.    The authors tried to evaluate the potential of BSFL frass as organic fertilizer. However, some critical indicators were missing, which severely hindering the validity and reliability of this evaluation. Besides the indicators mentioned in this manuscript, seed germination test, biochemical oxygen demand (BOD), chemical oxygen demand (COD), humic acid content, et al., especially the former one of which is critical to evaluate the biosafety of organic fertilizer. The authors are strongly suggested to supply these indicators, which are critical to construct the reliable evaluation.

4.    “Materials and Methods”: As the physicochemical (such as moisture, pH, electrical conductivity (EC)) and nutritional (such as total organic carbon content(TOC), total Kjeldahl Nitrogen (TKN), C:N ratio, et al) properties are critical for BSFL growth, the authors are suggested to add these information.

5.    “Figures”: Improve the figures’ clarity.

6.    Figure 1: As usual, the full name should be described when using the abbreviation for the first time in the figure legends, so the "BF-rOS", "EM" and "HIL/BSFL" should be supplemented with their full names.

7.    Line 191-192 and Table 1: I wonder the way to calculate the bioconversion efficiency of BF-rOS/FW mixture using BSFL in this manuscript, as it is not clarified clearly in the related method description in “2.4. Analysis of bioconversion efficiency of HIL”. According to Table 1, it seems like that the author calculate the bioconversion efficiency by dividing the residue weight by the BF-rOS weight, rather than the BF-rOS/FW mixture weight. However, the latter one is correct, which is suggested for correcting the bioconversion efficiency calculation. In addition, the bioconversion efficiency values and the standard limits of all indicators were suggested in adding into Table 1.

8.    I wonder why there is no statistic analysis in Table 2 to Table 5. In addition, the title of Table 5 is lost.

9.    References: The format of the text must be unified. For example, the first letters of all words of the journal titles should be capitalized.

Author Response

Thank you for your comments and suggestions. We have addressed your feedback below and hope you find our responses satisfactory.

This manuscript entitled “Resource utilization of residual organic sludge generated from bioenergy facilities using Hermetia illucens larvae” presents an important investigation of the bioconvertion of residual organic sludge generated from bioenergy facilities (BF-rOS) using black soldier fly larvae (BSFL) to generate the edible insect-sourced nutrition and organic fertilizers. The author not only conformed the BSFL could efficiently bioconvert the BF-rOS mixed with food waste (FW), further optimized the most efficient ratio of BF-rOS and FW mixtures, but also tried to evaluated the potential of insect biomass as animal feeding nutrition and frass as organic fertilizers. These findings provide a new efficient way to treat the BF-rOS for recycling this organic waste for fertilizer production and animal feeding. The manuscript content is logic, the results are clearly described, discussion is sufficient, and the writing is well readable. However, the deficiencies in method description, result analysis, figures, tables, are especially in the indicator adequacy to support the conclusion, were also found as described as followed:

  1. In most cases, the larvae of black soldier fly are abbreviated as “BSFL”, which is also suggested in this manuscript.

: Thanks for your comment. We changed the abbreviation of Hermetia illucens larvae to “BSFL”.

  1. Why selecting food wastes as the supplementary materials to improve the bioconversion of BF-rOS using black soldier fly larvae (BSFL)? Beside food wastes, many organic wastes, such as pig and chicken manures, are also good materials for BSFL conversion, so I suggest the authors to describe the reason for choosing FW.

: Thanks for your comments. As you commented, pig and chicken manures are excellent substances for BSFL conversion. However, in Korea, BSFL reared on these manures cannot be utilized as feed materials or additives. BSFL fed on food wastes, however, can be used as feed materials and additives. Therefore, we utilized food wastes as supplementary materials to improve the bioconversion of BF-rOS and to evaluate the possibility of BSFL fed on BF-rOS as a feed material or additive.

  1. The authors tried to evaluate the potential of BSFL frass as organic fertilizer. However, some critical indicators were missing, which severely hindering the validity and reliability of this evaluation. Besides the indicators mentioned in this manuscript, seed germination test, biochemical oxygen demand (BOD), chemical oxygen demand (COD), humic acid content, et al., especially the former one of which is critical to evaluate the biosafety of organic fertilizer. The authors are strongly suggested to supply these indicators, which are critical to construct the reliable evaluation.

: Thank you for your insightful comments. The primary objective of this investigation was to explore an environmentally friendly method for treating BF-rOS using BSFL and to evaluate the industrial potential of the resulting BSFL frass and larvae.

While we acknowledge the importance of the additional indicators you mentioned, such as seed germination test, BOD, COD and humic acid content, this investigation was specifically focused on the initial assessment of the nutrient content BSFL frass in accordance with the standards set by the Fertilizer Control Act on the Republic of Korea. This was done to establish a baseline for its potential use as a fertilizer ingredient.

The seed germination test is crucial for evaluating the biosafety, however, due to the scope and resource for our current study, we could not include these additional tests. We plan to address these important indicators in further research for a more comprehensive evaluation of BSFL frass as an organic fertilizer.

  1. “Materials and Methods”: As the physicochemical (such as moisture, pH, electrical conductivity (EC)) and nutritional (such as total organic carbon content (TOC), total Kjeldahl Nitrogen (TKN), C:N ratio, et al) properties are critical for BSFL growth, the authors are suggested to add these information.

: Thanks for your comments to include physiochemical and nutritional properties critical for BSFL growth. However, the primary objective of this study was to explore the environmentally friendly treatment of BF-rOS using BSFL and to evaluate the potential industrial application of the resulting BSFL frass and larvae. As such, our focus was on the end products rather than optimizing the rearing condition for BSFL. Further investigation will address the detailed growth conditions of BSFL to complement our findings.

  1. “Figures”: Improve the figures’ clarity.

: Thanks for your comments. All figures in the manuscript were originally created at 600 dpi resolution. We suspect that the resolution of the figures has decreased during the file format conversion. Therefore, we have attached the original photographs as separate files.

  1. Figure 1: As usual, the full name should be described when using the abbreviation for the first time in the figure legends, so the "BF-rOS", "EM" and "HIL/BSFL" should be supplemented with their full names.

: Thanks for your comments. As you mentioned we corrected the abbreviations to their full names.

  1. Line 191-192 and Table 1: I wonder the way to calculate the bioconversion efficiency of BF-rOS/FW mixture using BSFL in this manuscript, as it is not clarified clearly in the related method description in “2.4. Analysis of bioconversion efficiency of HIL”. According to Table 1, it seems like that the author calculate the bioconversion efficiency by dividing the residue weight by the BF-rOS weight, rather than the BF-rOS/FW mixture weight. However, the latter one is correct, which is suggested for correcting the bioconversion efficiency calculation. In addition, the bioconversion efficiency values and the standard limits of all indicators were suggested in adding into Table 1.

: Thanks for your comments. We have revised the calculation method for the bioconversion efficiency of the BF-rOS/FW mixture using BSFL as suggested. The calculation method was expressed in “2.4 Analysis of bioconversion efficiency of BFSL”. In addition, we added the revised bioconversion efficiency values [mentioned as Relative residual reduction (%)] and the standard limits for all indicators to Table 1.

  1. I wonder why there is no statistic analysis in Table 2 to Table 5. In addition, the title of Table 5 is lost.

: Thanks for your comment. We did not perform statistical analysis for Table 2 to 5 because all samples from the same experimental group were pooled for analysis. The analysis of BSFL frass was done on pooled samples due to the limitation of analytical costs. Similarly, the compositional analysis of BSFL was conducted on pooled samples because the total weight of BSFL fed BF-rOS or BF-rOS/FW (90:10) was too low for independent analysis.

In addition, we removed Table 5 as aflatoxins were not detected in any samples. However, the results of aflatoxin analysis are still presented in the manuscript as “No aflatoxins were detected in the BSFL (data not shown)”.

  1. References: The format of the text must be unified. For example, the first letters of all words of the journal titles should be capitalized.

: Thanks for your comment. We checked all references and corrected the format. It appears that the issues may have arisen due to system problem within the software (EndNote 20). Although we have mad corrections, there is a possibility that the corrected references could revert to their original state during the file save or modification process. To mitigate this potential issue, we have attached the corrected content as a PDF file. 

Reviewer 2 Report

Comments and Suggestions for Authors

This manuscript presents the study on the potential of residual organic sludge as black soldier fly (Hermetia illucens) larvae feed. Numerous trials were performed using sludge alone and mixed with different proportions of food waste, with and without the addition of effective microorganisms. The authors tested the BSFL growth, bioconversion efficiency, composition of frass, nutritional composition of BSFL, and presence of heavy metals and aflatoxins.

While the topic of the study is important and the results are potentially interesting, in my view, the manuscript should be improved before publication following the points below:

Some information on the composition of the organic sludge should be added to the Introduction section

Materials and Methods section should be expanded. For example, there is no information about the number of replicates per treatment (this is also mostly omitted in the results). Methods often refer to the Korean national standards that are not easily accessible to most readers. Briefly describing the methods or adding references is necessary. A reference for the “effective microorganism” should also be added. How many times were the larvae fed in 14 days?

Sawdust was added to the BSFL substrates to adjust moisture content. It is an inert substance in the sense that it can’t be processed by BSFL. Therefore, it will significantly influence the amounts of residue (frass). This was not mentioned neither in Results nor in the Discussion. Furthermore, comparing the weight of residues with different water contents is misleading. It would be better to compare dry weight (which should be possible to calculate from your data).

The Discussion section should present the results obtained in this study in the context of what is already known on the subject rather than just rephrasing the results. Some similar or relevant studies are not cited, e.g. https://www.mdpi.com/2305-6304/12/6/414, https://doi.org/10.1016/j.jclepro.2018.10.017, https://doi.org/10.1016/j.jenvman.2018.03.122.

Figures are blurry and difficult to see.

Lines 66 and 77

The term “eco-friendly“ sounds too colloquial  to me. I would suggest substituting it with sustainable, environmentally-friendly, or similar.

Lines 67-68

“… require no rearing cost…”

Perhaps you mean no feed cost? Although the BSF substrate might be cheap or free, there are still costs for equipment, labour, utilities, etc.

Lines 85 and 86

Please add more information about how the sludge was produced.

Figure 1.

What is the control?

Lines 171 and 172

“…supplementation of EM with the mixture…”

This should be “supplementation of the mixture with EM” or “addition of EM to the mixture”.

Figure 2.

For better understanding and consistency, OS should be replaced with BF-rOS. Including groups fed 100% BF-rOS would be interesting.

Line 199-212

For the evaluation of organic fertiliser, it would be important to test for other more relevant plant nutrients, such as K, P, different forms of N, Ca, etc.

Table 2

Is BF-rOS untreated sludge? Why does it have only 19% moisture?

Table 5. can be removed from the manuscript. Simply stating that aflatoxins were not detected in any of the samples is sufficient.

Author Response

Thank you for your comments and suggestions. We have addressed your feedback below and hope you find our responses satisfactory.

This manuscript presents the study on the potential of residual organic sludge as black soldier fly (Hermetia illucens) larvae feed. Numerous trials were performed using sludge alone and mixed with different proportions of food waste, with and without the addition of effective microorganisms. The authors tested the BSFL growth, bioconversion efficiency, composition of frass, nutritional composition of BSFL, and presence of heavy metals and aflatoxins.

While the topic of the study is important and the results are potentially interesting, in my view, the manuscript should be improved before publication following the points below:

  1. Some information on the composition of the organic sludge should be added to the Introduction section

: Thanks for your valuable feedback. We acknowledge the importance of providing such details to enhance the context and understanding of our investigation. However, we conducted the experiments using BF-rOS and food waste provided to us without detailed composition information. The investigation was commissioned by an external organization, and the samples were supplied directly by them. As a result, we did not have access to the specific compositional data of the organic sludge used in this investigation.

  1. Residual organic sludge generated from bioenergy facilities (BF-rOS) can cause water and air pollution and serve as a transmission route for pathogens

→ Residual organic sludge generated from bioenergy facilities (BF-rOS) can cause water and air pollution and serve as a transmission route for pathogens because it retains moisture, organic matter (such as protein, carbohydrate and lipid), minerals, and pathogens

  1. Materials and Methods section should be expanded. For example, there is no information about the number of replicates per treatment (this is also mostly omitted in the results). Methods often refer to the Korean national standards that are not easily accessible to most readers. Briefly describing the methods or adding references is necessary. A reference for the “effective microorganism” should also be added. How many times were the larvae fed in 14 days?

: Thanks for your comments. As you commented, we agree that Materials and Methods section are short and not easily accessible. Therefore, we described the methods in detail and added references for the “effective microorganism”. In addition, BSFL were initially fed once at the start of the rearing period. Therefore, we did not mention the frequency of feeding.

  1. Sawdust was added to the BSFL substrates to adjust moisture content. It is an inert substance in the sense that it can’t be processed by BSFL. Therefore, it will significantly influence the amounts of residue (frass). This was not mentioned neither in Results nor in the Discussion. Furthermore, comparing the weight of residues with different water contents is misleading. It would be better to compare dry weight (which should be possible to calculate from your data).

: Thanks for your comments. As you commented, sawdust can affect to the amount of residue of BR-rOS and BF-rOS/FW mixtures. Therefore, we indicated the potential impact of sawdust in “3.3 Bioconversion efficiency of BF-rOS and BF-rOS/FW mixtures by BSFL”.

In addition, all the residual matter was weighed after 14-day maturation as mentioned in ‘2.4. Analysis of bioconversion efficiency of BSFL’. During this process, all residual matters dried. We used bioconversion efficiency instead of the weight of residual matter to avoid potential confusion in understanding the bioconversion efficiency of BSFL. The bioconversion efficiency was calculated using the formulas presented in ‘2.4. Analysis of bioconversion efficiency of BSFL’. 

  1. The Discussion section should present the results obtained in this study in the context of what is already known on the subject rather than just rephrasing the results. Some similar or relevant studies are not cited, e.g. https://www.mdpi.com/2305-6304/12/6/414, https://doi.org/10.1016/j.jclepro.2018.10.017, https://doi.org/10.1016/j.jenvman.2018.03.122.

 : We acknowledge and appreciate your constructive feedback. Consequently, we have revised the discussion section by adding additional references, including those you recommended. We hope the discussion meets your expectations.  

  1. Figures are blurry and difficult to see.

 : Thanks for your comments. All figures in the manuscript were originally created at 600 dpi resolution. We suspect that the resolution of the figures has decreased during the file format conversion. Therefore, we have attached the original photographs as separate files.

  1. Lines 66 and 77

The term “eco-friendly“ sounds too colloquial  to me. I would suggest substituting it with sustainable, environmentally-friendly, or similar.

 : Thanks for your comment. We corrected “eco-friendly” to “environmentally-friendly”.

  1. Lines 67-68

“… require no rearing cost…”

Perhaps you mean no feed cost? Although the BSF substrate might be cheap or free, there are still costs for equipment, labour, utilities, etc.

 : Thanks for your comment. We did not consider equipment, labour, utilities, etc. We agree with your suggestion and have revised “require no rearing cost” to “require low rearing cost”.

  1. Lines 85 and 86

Please add more information about how the sludge was produced.

 : Thanks for your comment. We added detailed information about how the sludge was produced.

  1. Figure 1.

What is the control?
: Thanks for your comment. We missed mentioning what the control is. The control is “food waste”. We mentioned that food waste served as a positive control in Figure 1.

  1. Lines 171 and 172

“…supplementation of EM with the mixture…”

This should be “supplementation of the mixture with EM” or “addition of EM to the mixture”.

 : Thanks for your comment. We corrected “supplementation of EM with the mixture” to “supplementation of the mixture with EM”.

  1. Figure 2.

For better understanding and consistency, OS should be replaced with BF-rOS. Including groups fed 100% BF-rOS would be interesting.

 : Thanks for your comment. We replaced ‘OS’ with ‘BF-rOS’ and added a 100% BF-rOS group in the figure.

  1. Line 199-212

For the evaluation of organic fertiliser, it would be important to test for other more relevant plant nutrients, such as K, P, different forms of N, Ca, etc.

 : Thanks for your insightful comments regarding the evaluation of organic fertilizer. We acknowledge the importance of testing for various essential plant nutrient, such as K, P, different forms of N, Ca, etc. However, we did not perform these specific tests for the following reasons:

  1. Regulatory compliance: The Republic of Korea’s government has authorized BSFS frass as an organic fertilizer. The regulatory requirements do not mandate testing for the aforementioned elements (K, P, different forms of N, Ca) for approval and certification

  1. Experimental protocol: All experiments conducted in our study strictly adhered to the standards and guidelines established by Korean government. These standards are designed to ensure the safety and efficacy of organic fertilizer without requiring the specified elemental tests.

We also understand that including these tests could provide a more comprehensive evaluation of the fertilizer’s potential. However, our study focused on meeting the current regulatory standards in Korea, which we believe are sufficient for assessing the fertilizer’s suitability within the context of our national framework.

We hope this explanation clarifies our methodology and the rationale behind it.

  1. Table 2

Is BF-rOS untreated sludge? Why does it have only 19% moisture?

 : Thanks for your comments. Yes. This is untreated BF-rOS. However, as mentioned 2.5 Analysis of component content of BSFL frass, BF-rOS was also maturated for four weeks. During the maturation, the moisture originally present in BF-rOS decreased by up to 19% due to evaporation.

  1. Table 5. can be removed from the manuscript. Simply stating that aflatoxins were not detected in any of the samples is sufficient.

: Thanks for your comment., We removed Table 5.

Round 2

Reviewer 1 Report

Comments and Suggestions for Authors

The authors have appropriately responded and modified the manuscript according to my comments. After two additional minor revisions, I suggest accepting this manuscript:

1.       In the abstract, the recent results supplied in this manuscript did not fully support the conclusion in the last sentence, which was suggested being changed into “In conclusion, this investigation demonstrates that BF-rOS could be recycled for fertilizer production by mixing it with FW to feed BSFL, which generating the valuable insect biomass as potential nutrition for animal feeding”.

2.       In Figure 1B, the values of EM concentrations could not be easily identified, which were suggested to be rotated with specific angle rather than being horizontal.

Author Response

Thank you for your comments and suggestions. We have addressed your feedback below and hope you find our responses satisfactory.

The authors have appropriately responded and modified the manuscript according to my comments. After two additional minor revisions, I suggest accepting this manuscript:

  1. In the abstract, the recent results supplied in this manuscript did not fully support the conclusion in the last sentence, which was suggested being changed into “In conclusion, this investigation demonstrates that BF-rOS could be recycled for fertilizer production by mixing it with FW to feed BSFL, which generating the valuable insect biomass as potential nutrition for animal feeding”.

: Thanks for your kind suggestion. We acknowledge your suggestion and have accordingly revised the abstract as per your recommendation.

  1. In Figure 1B, the values of EM concentrations could not be easily identified, which were suggested to be rotated with specific angle rather than being horizontal.

: Thanks for your kind suggestion. We acknowledge your suggestion and have accordingly revised the Figure as per your recommendation. We have attached the revised figure file.

Reviewer 2 Report

Comments and Suggestions for Authors

The term "eco-friendly" still appears in the manuscript.

Author Response

Thank you for your comments. We have addressed your feedback below and hope you find our responses satisfactory.

The term "eco-friendly" still appears in the manuscript.

: Thanks for your kind comment. We corrected “eco-friendly” to “environmentally-friendly”.